# αV-Integrin-Dependent Inhibition of Glioblastoma Cell Migration, Invasion and Vasculogenic Mimicry by the uPAcyclin Decapeptide

**DOI:** 10.3390/cancers15194775

**Published:** 2023-09-28

**Authors:** Paola Franco, Iolanda Camerino, Francesco Merlino, Margherita D’Angelo, Amelia Cimmino, Alfonso Carotenuto, Luca Colucci-D’Amato, Maria Patrizia Stoppelli

**Affiliations:** 1Institute of Genetics and Biophysics “A. Buzzati Traverso” (IGB-ABT), National Research Council of Italy, 80131 Naples, Italy; paola.franco@igb.cnr.it (P.F.); iolanda.camerino@unicampania.it (I.C.); meghyd@icloud.com (M.D.); amelia.cimmino@igb.cnr.it (A.C.); 2Department of Environmental, Biological and Pharmaceutical Sciences and Technologies, University of Campania “Luigi Vanvitelli”, 81100 Caserta, Italy; 3Department of Pharmacy, University of Naples ‘Federico II’, 80131 Naples, Italy; francesco.merlino@unina.it (F.M.); alfocaro@unina.it (A.C.); 4Department of Experimental Medicine, University of Campania Luigi Vanvitelli, 81100 Naples, Italy; 5InterUniversity Center for Research in Neurosciences (CIRN), 80131 Naples, Italy; 6UniCamillus—Saint Camillus International University of Health Sciences, 00131 Rome, Italy

**Keywords:** glioblastoma (GBM), vasculogenic mimicry (VM), cell migration and invasion, anti-cancer peptides, uPAcyclin

## Abstract

**Simple Summary:**

Glioblastoma is the most devastating and widespread primary central nervous system tumor and there is a compelling need for innovative and effective therapeutic strategies. The remarkable vascularization sustaining this malignancy occurs through different mechanisms, including vasculogenic mimicry, i.e., the non-endothelial formation of new vessels, that is a promising therapeutic target. This study provides evidence that the recently described anti-invasive uPAcyclin decapeptide is remarkably active in reducing migration, invasion, and vasculogenic mimicry formation by human glioblastoma cells. These findings uncover novel uPAcyclin activities and provide a strong rationale for further clinical studies.

**Abstract:**

Among the deadliest human cancers is glioblastoma (GBM) for which new treatment approaches are urgently needed. Here, the effects of the cyclic decapeptide, uPAcyclin, are investigated using the U87-MG, U251-MG, and U138-MG human GBM and C6 rat cell models. All GBM cells express the αV-integrin subunit, the target of uPAcyclin, and bind specifically to nanomolar concentrations of the decapeptide. Although peptide exposure affects neither viability nor cell proliferation rate, nanomolar concentrations of uPAcyclin markedly inhibit the directional migration and matrix invasion of all GBM cells, in a concentration- and αV-dependent manner. Moreover, wound healing rate closure of U87-MG and C6 rat glioma cells is reduced by 50% and time-lapse videomicroscopy studies show that the formation of vascular-like structures by U87-MG in three-dimensional matrix cultures is markedly inhibited by uPAcyclin. A strong reduction in the branching point numbers of the U87-MG, C6, and U251-MG cell lines undergoing vasculogenic mimicry, in the presence of nanomolar peptide concentrations, was observed. Lysates from matrix-recovered uPAcyclin-exposed cells exhibit a reduced expression of VE-cadherin, a prominent factor in the acquisition of vascular-like structures. In conclusion, these results indicate that uPAcyclin is a promising candidate to counteract the formation of new vessels in novel targeted anti-GBM therapies.

## 1. Introduction

Glioblastoma (GBM) is recognized as the most aggressive form of neoplasia of the central nervous system, arising from glia or their precursor cells [1,2]. Although several treatment options are available, including surgery, along with radio- and chemotherapy, GBM is characterized by a very poor prognosis and life expectancy, with a median survival time ranging from 12 to 15 months. The treatment of GBM is based on a few conventional anti-cancer drugs, like temozolomide (TMZ) and cisplatin, two largely used DNA alkylating agents. However, most GBMs are not responsive to TMZ treatment, due to the high expression of *O*^6^-methylguanine-DNA methyltransferase (MGMT) [3]. Furthermore, following exposure to TMZ, a significant increase in the cancer stem cell (CSC) population, with an ability of self-renewal, tumorigenicity, as well as resistance to radio- and chemotherapy, has been observed [4].

One of the hallmarks of advanced GBM is a remarkable neovascularization, promoted by several mechanisms, including the recruitment of endothelial progenitor cells [5], vessel co-option [6], intussusceptive angiogenesis [7], and vasculogenic mimicry (VM).

Although anti-angiogenic therapies are a promising option in GBM, they are often transiently effective, as cells acquire resistance by activating alternative vascularization pathways, like VM [8]. Noticeably, as first discovered in melanomas, tumor cells undergoing VM form clearcut vascular-like structures, expressing endothelial associated genes [9]. Also, GBM cells form VM, providing a blood supply system in fluid-conducting channels to carry nutrients and oxygen, often associated with an aggressive phenotype [9]. Among VM markers are vascular endothelial cadherin (VE-cadherin), EphA2, MT1-MMP, and laminin-5, in the absence of the endothelial marker CD31 [9]. Immunohistochemistry studies on human surgical samples confirmed the occurrence of cell-lined blood vessels inside the GBM tumor, with a functional basement membrane but independent of normal endothelial and mural cells [10,11]. In GBM, the subset of self-renewing CSC, possibly derived from mutated adult neural stem cells or from oligodendrocyte progenitors, may transdifferentiate into endothelial-like cells, forming fluid-conducting vascular and pericyte-like structures, thus supporting tumor development [8,12,13]. Regardless of the origin, the mechanisms underlying VM are currently being investigated and regarded as targets in novel anti-GBM pharmacological interventions. As a matter of fact, anti-angiogenic therapies are a promising option in GBM, but they are often transiently effective, as cells acquire resistance by activating alternative vascularization pathways, like VM [14,15].

Another relevant feature of GBMs, leading to a limited efficacy of surgical resection, is the extensive infiltration/invasion into the adjacent normal brain tissues, likely occurring through the multiple interactions of GBM cells with the extracellular matrix (ECM) components and microenvironmental cells [16]. The modulation of matrix adhesion, that is a prerequisite for efficient migration and invasion, is mainly exploited by the integrin receptors, that can also signal in a ligand-independent manner [17]. In particular, the αV of the vitronectin receptor heterodimerizes with the overexpressed β3, β5, β6, and β8 subunits in primary colorectal, breast, bladder, lung, melanoma, and kidney tumors and in the corresponding brain metastases [18]. In particular, integrin-αV (ITGAV) is known to be involved in the migration of glioma cells and is considered to greatly contribute to cell invasion [19]. Relevant interactions of integrin αvβ3 with extracellular matrix proteins result in enhanced focal adhesion kinase (FAK) signaling and metabolic reprogramming, thus promoting glioblastoma cell growth, migration, and invasion [20].

Interestingly, the expression of integrin αVβ3 provides a means to escape senescence during GBM tumor development, through activation of the cytoskeletal-associated PAK4 kinase [21]. In a recent study on 132 GBM patients, the overexpression of αV is associated with a poor prognosis, together with VEGFR3, murine double minute 2 (MDM2), and matrix metallopeptidase 2 (MMP2) [22]. Both αVβ5 and αVβ3 integrins are expressed at the tumor–normal tissue interface and a variety of compounds targeting integrins have been developed. Although not all results of the clinical trials have been encouraging, this approach still holds [19].

Given the poor survival with the currently approved treatments for GBMs, new and effective therapeutic strategies are needed. In previous studies, directed to the identification of anti-GBM active compounds, we tested histone deacetylase inhibitors (HDACi), inhibiting histone deacetylases (HDAC) activity and reducing cell growth, migration, and invasion in other tumors. HDACi are potent inhibitors of vascular-like structure formation by U87-MG GBM cells, suggesting their use in novel anti-GBM strategies [23,24]. Among the candidate compounds, peptide-based therapeutics show a highly specific targeting capacity and are associated with reduced side effects with respect to conventional anticancer therapies: in general, they could be used to target specific receptors, to deliver drugs, as ligand antagonists, in monotherapy, or in combination therapies.

Thus, in the present work, we investigated the effects of the anti-migratory uPAcyclin decapeptide, binding to the αV integrin subunit [25]. This cyclic decapeptide is derived from a non-catalytic region of the human urokinase (uPA) plasminogen activator. The growth factor domain (GFD, residues 1–49) and the connecting peptide region (CP, residues 132–158) confer to the full uPA a clearcut motogen activity. While the GFD binds to the urokinase receptor (uPAR), the CP region binds to the αV integrin subunit [26]. Phosphorylation studies revealed that Ser^138/139^, included in the CP region, are crucial to the migratory properties of uPA [27,28]. The knowledge derived from these studies led to the design of peptides retaining modulatory properties of cell migration [29]. The occurrence of a turn structure around the 140–143 segment in the CP-derived anti-migratory peptides suggested the design of two novel peptides corresponding to the N-terminal region of CP, with the S138E substitution, allowing the E138-K145 side-chain-to-side-chain contact, thus stabilizing the putative bioactive conformation. In particular, the novel uPAcyclin cyclic decapeptide (Ac-K^1^P[ESPPEELK^10^]-NH_2_) binds to the αV-integrin and prevent lung metastases in mice injected with HT1080 fibrosarcoma cells [29]. Furthermore, uPAcyclin inhibited the pro-invasive ability of cancer-associated fibroblasts (CAFs) from breast cancer patients, leading to a partial reversion of the CAF phenotype [30].

Here, conformational and functional analyses of uPAcyclin in the GBM context are presented. Moreover, using four cell lines (U87-MG, U251-MG, and U138-MG human GBM and C6 rat glioma cells), we have evaluated the ability of this cyclic decapeptide to inhibit migration and invasion, as well as the formation of vascular-like structures. Interestingly, uPAcyclin, while not affecting cell proliferation and survival, was found to be a very effective inhibitor of migration, invasion, and VM formation with a remarkable activity in the low nanomolar range.

## 2. Materials and Methods

### 2.1. Cell Lines and Culture Conditions

The U87-MG, U251-MG, and U138-MG human glioblastoma cell lines, C6 rat glioma, and the embryonic kidney HEK-293 and HEK-293/αV (HEK-293 overexpressing αV) cell lines were cultured in Dulbecco’s Modified Eagle Medium (DMEM, Life Technologies, Paisley, UK) supplemented with 10% fetal bovine serum (FBS, Life Technologies, Paisley, UK), in the presence of 100 U/mL of Na-penicillin and 100 µg/mL of streptomycin sulfate. The U87-MG and U251-MG cell lines were purchased from ECACC (European Collection of Authenticated Cell Cultures, Salisbury, UK) and ATCC (American Type Culture Collection, Manassas, VA, USA), respectively, whereas the C6, HEK-293, and U138-MG cells were obtained from ICLC (Interlab Cell Line Collection, National Institute for Cancer Research, Genoa, Italy). The HEK-293/αV stable transfectants have been described elsewhere [25].

### 2.2. Peptide Synthesis

The cyclic decapeptide uPAcyclin and the scrambled control peptide (S-uPAcyclin, Ac-PS[EELKPEPK]-NH_2_) were synthesized, as previously described [25,30]; uPAcyclin and S-uPAcyclin N-terminal ends were acetylated, whereas FITC-uPAcyclin was conjugated to fluorescein, as reported [31]. Details are reported in Appendix A.

### 2.3. NMR Conformational Analysis

NMR samples were prepared by dissolving uPAcyclin in 0.60 mL of DMSO-d6, to obtain a 1 mM final concentration. NMR spectra were recorded on a Bruker Avance NEO 600 MHz spectrometer equipped with a z-gradient 5 mm triple-resonance probe head at 25 °C. One-dimensional (1D) NMR spectra were recorded in the Fourier mode with quadrature detection. Two-dimensional DQF-COSY [32,33], TOCSY [34], and NOESY [35] spectra were recorded in the phase-sensitive mode using the method from States et al. [36]. The data block sizes were 2048 addresses in t2 and 512 equidistant t1 values. Before Fourier transformation, the time domain data matrices were multiplied by shifted sin2 functions in both dimensions. A mixing time of 80 ms was used for the TOCSY experiments. NOESY experiments were run with a 100 ms mixing time. The qualitative and quantitative analyses of DQF-COSY, TOCSY, and NOESY spectra were obtained using the interactive program package XEASY [37]. Complete ^1^H NMR chemical shift assignments were effectively achieved for uPAcyclin, according to the Wüthrich procedure [38], via the application of DQF-COSY, TOCSY, and NOESY experiments (Appendix A). The distance restraints were obtained from NOE intensities observed in the NOESY spectra (Appendix A). The NOESY cross peaks were converted into upper distance bounds making use of the CALIBA program featured into the program package CYANA [39]. Only constraints derived from NOE were employed in the annealing procedures. An ensemble of 100 structures was generated with the simulated annealing of the program CYANA. Among these, 7 structures were chosen, based on the best fit of the interproton distances and NOE-derived distances. The chosen structures were refined through successive steps of restrained and unrestrained energy minimization calculations with the Discover algorithm (Accelrys, San Diego, CA, USA) and the consistent valence force field [40]. The minimization lowered the total energy of the structures. Moreover, no residue was found in the disallowed region of the Ramachandran plot. The InsightII program (Accelrys, San Diego, CA, USA) was employed for analyzing the final structures.

Since the vitronectin crystal structure is not available, we used the available 3D structure of fibronectin (the tenth type III RGD domain of wild-type fibronectin, pdb code 4MMX) to compare the structure of uPAcyclin with that of an RGD-containing physiologic ligand of the αV integrin. 

### 2.4. Cell Viability Assay

Cell viability was tested in DMEM/10% FBS, in the presence of 1 nM or 100 nM uPAcyclin, either by Trypan blue exclusion or by thiazolyl blue tetrazolium bromide (MTT) assays, as specified in the figure legends. For the Trypan blue assay, 4 × 10^4^ U87-MG, U251-MG, U138-MG, and C6 cells/sample were grown in 12-well plates, and followed for 24, 48 and 72 h. As a negative control, the cells were grown in a serum-free medium. Then, the cells were detached, stained with a 0.2% (*w*/*v*) Trypan blue solution, counted using a Burker chamber (T0 = 100%), and reported as number of viable cells/total.

For the MTT assay, 5 × 10^3^ U87-MG and C6 cells/well were grown in 96-well plates for 24 h and tested as previously described [24]. Briefly, MTT (Invitrogen-Life Technologies, Eugene, OR, USA) 5 mg/mL in a PBS solution was added to wells at a 10% final concentration, and incubated for 3 h at 37 °C, followed by the addition of stop solution (100 µL of DMSO/well) for 1 h at room temperature. Then, the optical densities were measured at 495 nm using the Microplate Reader (Synergy HT, Bio-Tek, Winoosky, VT, USA). The cell viability rate was calculated as a percentage by reporting the mean absorbance of treated groups to control groups, as follows: Cell viability rate (%) = (A_treated group_/A_control group_) × 100.

### 2.5. Binding Assay

The U87-MG, U251-MG, and U138-MG cell lines were harvested and acid-treated, as described [41]. Then, 2 × 10^6^ cells/sample were exposed for 30 min at 4 °C to uPAcyclin or S-uPAcyclin at the indicated concentrations, or with the following antibodies: VNR147 monoclonal anti-αV integrin (Chemicon Int. Inc., Temecula, CA, USA), polyclonal anti-αV (Millipore, Burlington, MA, USA), monoclonal anti-α2 (Chemicon Int. Inc., Temecula, CA, USA), polyclonal anti-actin (Sigma-Aldrich, St. Louis, MO, USA), monoclonal anti-GAPDH (Abcam, Cambridge, UK), and purified vitronectin (VN, Corning, Glendale, AZ, USA). The cells were subsequently incubated with 50 nM FITC-uPAcyclin for 2 h at 4 °C, as described [25]. At the end of incubation, the cells were washed and the cell surface-associated fluorescence was determined using a fluorimeter (VICTOR^TM^ X3-PerkinElmer, Waltham, MA, USA). All binding assays were performed three times, with duplicate samples, and analyzed as described in Section 2.11.

### 2.6. Protein Extraction and Western Blotting Analysis

Whole cells were lysed as described [42], and resolved by SDS-PAGE, followed by Western blotting with PVDF membranes (Millipore, Burlington, MA, USA). The filters were probed with the following antibodies: polyclonal anti-VE Cadherin (Bioss Antibodies, Woburn, MA, USA), polyclonal anti-αV (Millipore), polyclonal anti-GAPDH (Elabscience, Houston, TX, USA), and polyclonal anti-actin (Sigma-Aldrich, St. Louis, MO, USA). Then, goat anti-rabbit IgG-HRP (Sigma) or goat anti-mouse IgG-HRP (Santa Cruz Biotechnology, Dallas, TX, USA) were employed as secondary antibodies. The peroxidase activity was revealed with Immobilon^TM^ Western HPR Substrate (Millipore). Films were imaged using the CanoScan 4400F (Canon, Tokyo, Japan) at 300 dpi, with the Adobe Photoshop Creative Suite 2 or CS2, and bands were quantified with the ImageJ 1.52a software (NIH, Bethesda, MD, USA).

### 2.7. Migration and Invasion Assays

The migration and invasion assays were performed using Boyden chambers and 8 µm pore size PVP-free filters coated with 50 µg/mL of collagen type VI (Sigma), as previously described [42]. For the invasion assays, collagen-coated filters were further coated with 50 µg/mL Matrigel^TM^ (Sigma). Briefly, 11 × 10^4^ U87-MG, 4 × 10^4^ U251-MG, or 4 × 10^4^ U138-MG/sample were pre-treated for 40 min at 37 °C with the indicated effectors. Then, the cells were allowed to migrate for 3 h or to invade for 5 h at 37 °C toward 5% FBS in DMEM/0.1% BSA. At the end of both assays, the cells on the lower filter surface were fixed, stained with haematoxylin, and counted under an optical microscope (Zeiss Axiophot, Carl Zeiss, Milan, Italy) at 20× magnification. The migration and invasion experiments were conducted in triplicate for U87-MG and in duplicate for U251-MG and U138-MG. The migrated or invaded cells were reported as percentage of basal migration or invasion in the presence of DMEM/0.1% BSA, taken as 100%, and analyzed as reported in Section 2.11.

### 2.8. Wound Healing Assay

The wound healing assay or scratch test was performed as previously reported [25]. Briefly, 6 × 10^3^ U87-MG cells were seeded in multiwell-12 plates containing culture inserts for live cell analysis (Ibidi, Gmbh, Martinsried, Germany) and grown to confluence in DMEM/10% FBS for 24 h. After removing the inserts, the cells were pre-treated for 40 min at 37 °C with uPAcyclin or diluents at the indicated concentrations, and then incubated in DMEM/3% FBS for 24 h. The images were taken using the Inverted Microscope Leica DMI6000 DFC 402 (Leica Microsystems, Milan, Italy). A total of 10 × 10^4^ C6 cells were grown to confluence in 12-well plates, in the presence of DMEM/10% FBS. Then, the cell monolayers were wounded with a yellow tip, pre-incubated with uPAcyclin (10 pM and 100 nM) or diluents, and incubated in DMEM/2.5% FBS for 16 h. The images were obtained using the Inverted Axiovert 25 Microscope, equipped with the Zen 3.3 Software. To quantitate cell migration, a rectangle covering the wound edges is drawn on the T0 image and is further applied to the single photograms of each sample. The average margin distance is measured at three points and the results are expressed as percentage wound width of the T0 distance, taken as 100%.

### 2.9. Vasculogenic Mimicry Assay

The tube formation assay was performed in 96-well plates pre-coated with 50 µL/well of Geltrex^TM^ (Thermofisher, Scientific, Waltham, MA, USA), and polymerized for 30 min at 37 °C. Briefly, 3 × 10^4^ U87-MG, 1.5 × 10^4^ U251, and 1 × 10^4^ C6 cells/100µL/well were plated onto Geltrex^TM^ and monitored at 37 °C for the indicated times. The specific effectors were included at time 0. The formation of vascular-like structures by Geltrex^TM^-plated U87-MG and U251-MG cells was monitored for 24 h, whereas C6 cells were tested for 16 h. The evaluation of the branching point number, defined as the intersection of at least 3 tubes [24], was performed using the Inverted Microscope DMI Leica 6000 for the U87-MG and U251-MG cells and the Inverted Microscope Axiovert 25 for the C6 cells. At the indicated times, the number of “branching points” was assessed in 4/5 random fields using the ImageJ 1.52a Software.

### 2.10. Cells Recovery from 3D Matrix and Lysate Preparation

To analyze matrix-embedded U251-MG cells, cells undergoing VM for 24 h were extracted using a 2× mixture of Trypsin, Accutase, and Dispase (1:1:1), acting for 45 min at 37 °C. To evaluate the VE-cadherin expression, the U251-MG cells were harvested, centrifuged at 3000 rpm for 7 min at 4 °C, and then ice-lysed for 30 min in an appropriate volume of RIPA buffer (10 mM Tris HCl at pH 8, 140 mM NaCl, 0.5% NP-40, 1 mM EDTA, 1 mM EGTA), containing protease and phosphatase inhibitors (1X Complete TM protease inhibitor cocktail, Sigma), as previously described [42]. At the end of incubation, the lysates were centrifuged at 12,000 rpm for 15 min at 4 °C to pellet insoluble products. The supernatants were collected and the protein concentration was quantified using the Bradford assay (Biorad, Hercules, CA, USA).

### 2.11. Statistical Analyses

Data are expressed as mean ± standard deviation (SD), indicated by error bars. Student’s *t*-test was conducted to analyze differences between data sets, indicated in the figures with *p* values ≤ 0.05 (*), ≤0.005 (**), or ≤0.001 (***).

## 3. Results

### 3.1. NMR Conformational Analysis of uPAcyclin

Our previous work highlighted the multiple anti-cancer activities of the uPAcyclin decapeptide, including its anti-migratory and anti-invasive effects and its ability to interfere with the activation of primary breast CAFs. Also, this novel peptide markedly inhibits metastatic lung spreading in nude mice [25]. The primary sequence of this cyclic peptide, derived from the connecting peptide region (CP) of the serin protease urokinase (uPA), is reported in Figure 1A. As assessed by a previous conformational analysis of uPAcyclin performed in water solution, about half of the peptide molecules exhibit the Ser^4^-Pro^5^ amide bond in *cis* configuration [25].

However, most overlapping signals arising from the *trans* and *cis* sub-populations prevented the identification of the nuclear Overhauser effect (NOEs) for structure calculation. Therefore, uPAcyclin was investigated using the DMSO-d_6_ solvent. It turned out that, in this solvent, the Ser^4^-Pro^5^ amide bond was exclusively in the *cis* configuration. Simulated annealing calculations based on NMR-derived constraints (Appendix A) disclosed the structure depicted in Figure 1. The peptide shows a ɣ-turn around Pro^6^ flanked by two extended regions. Because uPAcyclin binds to αV integrin, the structure of this decapeptide has been compared to the integrin-binding region of fibronectin including the RGD sequence, that is, the main integrin-binding domain occurring in the extracellular matrix proteins. In Figure 1B, the 3D structure of uPAcyclin is superimposed to the relevant sub-region of fibronectin for direct comparison. It is evident that the pharmacophoric charged side chains of fibronectin cannot be efficiently overlapped with the putatively corresponding ones (Lys^1^ and Glu^7^ or Glu^8^) of uPAcyclin; this suggests that the binding of the decapeptide to integrin is likely to occur in a different, RGD-independent mode.

### 3.2. Specific Binding of FITC-uPAcyclin to U87-MG, U251-MG, and U138-MG GBM Cells

To investigate whether uPAcyclin could target GBM cells, we first checked the αV integrin expression of the GBM cell lines. To this end, total lysates from U87-MG, U251-MG, and U138-MG cells were compared to lysates of embryonic kidney HEK-293 cells overexpressing the αV-integrin subunit (HEK-293/αV) and to their parental HEK-293 by Western blotting (Figure 2A). Unlike the HEK-293 cell line, all three GBM cell lines and HEK-293/αV showed αV expression. Furthermore, the specific interaction of FITC-conjugated uPAcyclin with U87-MG, U251-MG, and U138-MG living cells was tested by a receptor binding assay, using HEK-293 and HEK-293/αV as controls (Figure 2B). This assay confirmed that FITC-uPAcyclin binds specifically to all three GBM cell lines and to the HEK-293/αV, whereas little interaction is detected with HEK-293 cells. A further binding assay with increasing concentrations of unlabeled uPAcyclin resulted in a progressively decreased FITC-uPAcyclin binding with a Kd_app_ in the low nanomolar range, showing a high-affinity interaction with U87-MG, U251-MG, and U138-MG cells (Figure 2C). Further tests show that S-uPAcyclin, carrying the same amino acids in a scrambled order, retains very little binding ability to U87-MG cells (Figure 2D). This may be due to the persisting interactions of critical amino acids shared between the sequences of uPAcyclin and S-uPAcyclin. As expected, the high-affinity interaction of FITC-uPAcyclin with U87-MG cells is inhibited by anti-αV polyclonal and monoclonal antibodies, and vitronectin, the last only at a high concentration, whereas anti-α2 integrin, anti-actin, or anti-GAPDH antibodies are ineffective (Figure 2D). These data confirm the occurrence of high-affinity, αV-specific binding of FITC-uPAcyclin to the GBM cell lines.

### 3.3. Inhibition of U87-MG, U251-MG, and U138 GBM Cell Migration and Invasion by uPAcyclin

Belli et al. [25] showed that the uPAcyclin peptide is a potent inhibitor of migration and invasion of breast cancer cell lines. Here, the αV-dependent inhibition of GBM cell migration was investigated by chemotactic assays in Boyden chamber devices in the presence of pico- and nanomolar concentrations of uPAcyclin. Unlike the S-uPAcyclin that is ineffective, 1 nM uPAcyclin caused a 70–80% inhibition of U87-MG, U251-MG, and U138-MG GBM cell migration, and 100 nM reduced migration to below background levels (Figure 3A–C).

These results are in agreement with the Kd_app_ in the low nanomolar range, resulting from the binding data shown in Figure 2C. In this assay, the S-uPAcyclin peptide is ineffective up to 100 nM (Figure 3A,B). If the GBM cell lines are previously exposed to polyclonal or monoclonal anti-αV antibodies, their directional migration is strongly inhibited, indicating the relevance of αV to migration in the GBM cellular model. In contrast, anti-α2 and anti-GAPDH antibodies are irrelevant.

Similar experiments were conducted to test cell invasion, following incubation of U87-MG, U251-MG, or U138-MG cells with 100 nM or 1 µM uPAcyclin or diluents, and allowing cells to invade in Boyden chambers for 5 h, in the presence of Matrigel. The assay was carried out in the presence of 5% FBS as a chemoattractant (Figure 4).

All GBM cells were invasive, unless pre-incubated with uPAcyclin; in particular, 100 nM peptide caused a 50–60% reduction in invasion, whereas 1 µM reduced invasion to basal levels (Figure 4A–C). The concentrations of peptide required for reducing invasion, definitely higher than those effective in migration, may be due to the prolonged time of the invasion assay, possibly influencing peptide stability. Furthermore, the invasion assays confirmed the relevance of αV’s modulatory role, as polyclonal and monoclonal anti-αV antibodies prevented U87-MG cell invasion whereas anti-actin did not (Figure 4A).

To rule out the possibility that the effects of uPAcyclin could be due to the inhibition of cell proliferation, cell growth in the presence of uPAcyclin was monitored for 72–96 h (Appendix A). U87-MG, U251-MG, and U138-MG cells were exposed to diluents or to serum in the absence or in the presence of uPAcyclin 1 nM or 100 nM, and cell viability was tested at the indicated times using Trypan blue (Appendix A–C) or MTT assays (Appendix A). The growth rates of all three cell lines were similar, regardless of the exposure to 1 nM or 100 nM uPAcyclin. All cell lines show serum- and time-dependent proliferation, that is unaffected by uPAcyclin, regardless of the concentration employed.

### 3.4. Effect of uPAcyclin in Wound Healing Assays of U87-MG and C6 Cells

To study the non-directional cell migration in culture, one option consists of the evaluation of migration after having wounded a cell monolayer. Afterwards, the filling of the empty area by migrating cells is monitored at specific times and the resulting images are visualized under an inverted microscope. Prior to the assay, cells are exposed to 10 pM or 100 nM uPAcyclin or diluents. As shown in Figure 5A, in the absence of treatment, cells migrate into the wound in a serum- and time-dependent manner. The quantitative evaluation shows that the distance between margins after 24 h is 15% of the initial wound length. In contrast, in the presence of 10 pM or 100 nM uPAcyclin, the residual space is 30% or 60% of the initial wound, respectively (Figure 5C). This shows a strong and concentration-dependent inhibition of wound closure by uPAcyclin.

Given the evolutionary conservation of integrin chains, we investigated whether this inhibitory effect could be extended to other non-human cell lines, like the C6 rat glioma cells. Upon exposure of the C6 wounded monolayers to 10 pM or 100 nM uPAcyclin, a dose-dependent inhibition of wound closure was also observed in this cell line (Figure 5B,C). Overall, the inhibitory peptide concentrations are in good agreement with those observed in the human GBM line directional migration assays (Figure 3). Importantly, proliferation and viability are not affected by uPAcyclin in any of the analyzed cell lines, as shown by the Trypan blue exclusion and MTT tests (Appendix A).

### 3.5. Vascular Mimicry Formation by U87-MG and U251-MG Glioblastoma Cell Lines

When U87-MG human glioblastoma cells are seeded onto culture dishes, they grow adherent to the surface without forming any particular structure (Figure 6). When cultivated under serum-free conditions in gelled extracellular matrices like Geltrex^TM^, U87-MG cells are able to form clearcut vascular-like structures in 8–24 h, developing from elongated cell bodies that connect to each other and form polygonal networks. Such vascular-like structures resemble those formed by human endothelial cells under similar experimental conditions [43,44]. Upon tube formation, U87-MG cells express VE-cadherin, a hallmark of VM [24]. Here, U87-MG cells were plated onto Geltrex^TM^ and the formation of round vascular-type structures was monitored over time (Figure 6A,B). The quantitation of the increased branching points was accomplished, taking into account the intersections involving at least three protrusions (Figure 6B,C). Cell exposure to the uPAcyclin decapeptide at 10 pM, 10 nM, and 100 nM leads to a concentration-dependent reduction in the branching point number (Figure 6B). Quantitative data, in the presence of uPAcyclin at 1 nM or 100 nM, show an approximately 60% reduction in branching points (Figure 6C).

An approximately 20% reduction is observed with the S-uPAcyclin, suggesting the persistence of low-affinity interactions with αV integrin. In an independent experiment, U87-MG cells were seeded on Geltrex^TM^ matrix in the absence of serum and VM formation was monitored for 14 h by time-lapse videomicroscopy. Cells were exposed either to diluents (Appendix A) or to 100 nM uPAcyclin (Appendix A). Representative videos are provided as Appendix A. It can be noticed that VM formation is inhibited and delayed in Appendix A, as compared to Appendix A.

Inhibition by uPAcyclin is observed also when seeding C6 cells under the conditions described in Figure 6 and monitoring them for VM formation. When exposed to 10 pM or 100 nM uPAcyclin, the branching point number is inhibited by 40% and 60%, respectively (Figure 7A,B).

In the following experiment, the expression of VE-cadherin, that is, a marker of vascular structure formation, is evaluated in the presence and in the absence of uPAcyclin. To this end, U251-MG GBM cells were seeded onto Geltrex^TM^ in the absence of serum and monitored for the formation of vascular-like structures. The cells started extending protrusions and forming contact structures after 8 h, and tube formation progressively increased after 24 h, whereas, in the presence of 1 nM or 100 nM uPAcyclin, the branching number per field decreased to 20% and 15%, respectively. This reveals a high sensitivity to the inhibition by uPAcyclin (Figure 8A,B). After 24 h, the expression of VE-cadherin was analyzed by extracting vascular-like cells from the matrix by enzymatic digestion. The recovered cells were lysed and subjected to Western blotting analysis. Cells undergoing VM showed the 130 KDa VE-cadherin band (Figure 8C, right panel). In contrast, when treated with uPAcyclin, the extent of VE-cadherin expression was markedly reduced (Figure 8C, left panel).

Overall, these data conclusively show that the uPAcyclin peptide is able to inhibit migration, invasion, and VM formation of GBM cells, as well as the expression of VE-cadherin by vascular-like GBM cells.

## 4. Discussion

Among new anti-cancer therapeutics, peptides show the potential for effective targeting associated with reduced side effects and drug resistance [45,46]. In this study, the anti-migratory, anti-invasive, and anti-vasculogenic structure formation activities of the urokinase-derived uPAcyclin decapeptide are presented. First, we assessed that this cyclic peptide binds with high affinity and specificity to the αV integrin subunit on the membrane of U87-MG, U251-MG, and U138-MG human GBM cells. Moreover, uPAcyclin is shown to be a potent inhibitor of GBM cell migration and invasion at nanomolar concentrations. In this regard, the role of αV integrin, the specific target of uPAcyclin, was assessed by testing anti-αV integrin antibodies on migration and invasion of GBM cells. These experiments indicated a crucial role of this integrin subunit in the modulation of GBM migration and invasion. Furthermore, the ability of human U87-MG and U251-MG glioblastoma and C6 rat glioma cells to form vascular-like structures is markedly inhibited by nanomolar concentrations of uPAcyclin. Finally, impaired vascular formation by U251-MG cells is associated with decreased levels of VE-cadherin, a relevant factor for the formation of vascular-like structures.

The binding specificity of the decapeptide to the αV integrin subunit was confirmed by using HEK-293 overexpressing αV integrin, compared to parental cells. The former cells exhibited an increased ability to bind uPAcyclin. This specificity was further corroborated by the findings that anti-αV antibodies and vitronectin inhibited uPAcyclin binding to intact GBM cells (Figure 2).

The data presented here not only indicate the inhibitory potential of uPAcyclin, but also uncover a relevant role for αV-integrin in the modulation of migration, invasion, and VM formation in GBM cells. This well agrees with the reported overexpression and functional relevance for αV integrin in the GBM malignancy.

It is noteworthy that the inhibitory effect of uPAcyclin is also exerted on the αV-expressing U138-MG cell line, not undergoing VM, but sensitive to the anti-migratory and anti-invasive effect of the decapeptide. The expression of αVβ3 and αVβ5 integrins is up-regulated in different tumors, including gliomas [47]. Moreover, in brain tumors, ITGB8 and ITGAV integrins show the highest expression in primary proneural and mesenchymal gliomasphere cultures, while their interaction with the integrin-binding syalo-protein IBSP promote tumor cell migration. In this system, ITGAV knockdown reverses the pro-migratory effects of IBSP [48]. IBSP is an important motogen factor, known to interact with αVβ3 and αVβ5 through an internal RGD sequence, promoting cell proliferation and invasion, respectively [49]. It is worth noting that, from a therapeutic perspective, the restricted expression of αVβ3 in normal tissues confers to this integrin a high potential as a target for CAR-T-mediated immunotherapy of GBM tumors [50].

Since the identification of the integrin-binding domain (RGD) in fibronectin/vitronectin, RGD-containing peptides and derived compounds have been developed. Among these, cilengitide, a known αVβ3/αVβ5 integrin inhibitor, demonstrated a cytotoxic, anti-angiogenetic, and anti-invasive effect in GBM models [51]. Clinical trials with cilengitide in patients with advanced glioblastoma seemed to suggest a positive effect on overall survival [52]. In contrast, other trials have not shown cilengitide to be better in terms of progression-free survival or overall survival than the standard treatments [53]. Although the mechanistic issues have not been clarified, one of the reasons suggested to explain the failure of cilengitide in GBM patients is the detrimental stimulation of tumor angiogenesis at lower doses [54]. However, to our knowledge, the effect of cilengitide on the formation of vascular-like structures in culture has not been evaluated.

Integrin-targeting peptides have been proposed as drug conjugates in targeted drug delivery systems. For example, Zhan et al. reported that paclitaxel (PTX)-loaded cyclo-RGD enhanced PTX anti-tumor effects, both in vitro and in vivo in the orthotopic GBM model [55]. Among selected peptides used for GBM targeting, there is also Angiopep-2 which targets the low-density lipoprotein receptor related protein (LRP), highly expressed in GBM cells. In particular, Angiopep-2 was used to modify polyethylene glycol-based nanoparticles (PEG-NP) loaded with paclitaxel, and in vivo studies have shown that ANG-PEG-NP-PTX reduces tumor size two-fold as compared to Taxol and PEG-NP [56]. Another peptide-based strategy is to target key cellular signaling pathways involved in GBM, for example, the constitutive activation of the NF-kβ pathway that has been shown to drive a mesenchymal pattern of differentiation associated with shorter survival and radio-resistance [57]. Thus, Friedmann-Mezvinsky et al., using an RNA-Seq and bioinformatics analysis, identified genes enriched in GBM and designed a peptide denoted as NEMO-binding domain (NBD), which blocks the interaction of NEMO with the IKK (IkB)-kinase complex, thus inhibiting NF-kβ activity. This, in turn, resulted in slowed tumor growth both in mouse and in human GBM models, and extended survival time [58]. Similarly, it is likely that a number of “omic” approaches could be instrumental in identifying molecular targets involved in GBM transdifferentiation leading to vascular-like structures [59,60,61].

Unlike most peptide ligands of αV integrins containing the integrin-binding RGD sequence [62], the primary sequence of uPAcyclin does not contain any RGD sequence, suggesting a different binding mode. The possibility that Lys^1^ and Glu^7^ or Glu^8^ of uPAcyclin could mimic the Arg and Asp, respectively, in the RGD segment is unlikely. In fact, in all RGD-binding subtypes, except for α_IIb_β_3_, the guanidine group is bound via bifurcated salt bridges to the α-subunit and an amine is not recognized [63]. Moreover, considering the superimposition of the NMR-derived 3D structure of uPAcyclin and the available crystal structure of fibronectin [64], the pharmacophoric charged side chains of fibronectin cannot be efficiently overlapped with the corresponding ones of uPAcyclin (Figure 1B). Hence, the competition of FITC-uPAcyclin with vitronectin, observed at a concentration as high as 200 µM vitronectin (Figure 2D), may be due to sterical hindrance of a larger binding molecule. This supports the intriguing hypothesis that, unlike the classical RGD-containing ligands, uPAcyclin binds to αV integrin in a close allosteric site, thus affecting RGD ligand binding. However, this finding will need further investigation.

The sequence of uPAcyclin shares extensive similarities with the Å6 octapeptide (Ac-K^136^PSSPPEE^143^-NH_2_), corresponding to uPA residues 136–143, endowed with clearcut anti-angiogenetic and anti-metastatic activities in mice [65]. The Å6 was tested in gynecologic malignancies and phase I trials indicated that it was well tolerated. Further phase II studies indicated a delayed disease progression in the treated individuals [66,67].

New classes of chemotherapeutic agents or alternative methodologies that can overcome the limitations of current treatment options for GBM are being implemented. Novel drugs overcoming GBM drug resistance to TMZ, acquired through loss of the mismatch repair pathway, are being designed and tested at pre-clinical level, like the TMZ analogues that reduce tumor size and increase survival in mouse GBM xenografts [3]. Nevertheless, VM represents a new challenge to the therapy of GBM. Indeed, VM is associated with a more severe prognosis and is an important mechanism of anti-angiogenesis-targeted drug resistance [68,69]. VM involves extensive extracellular matrix remodeling with matrix metalloproteinases (MMP) playing a pivotal role [70]. In this regard, it is well known that matrix modeling also parallels cell migration and invasion. In turn, cell migration and invasion are prerequisites for VM [69]. It is worth noting that, in the experiments presented here, uPAcyclin has been shown to inhibit migration, invasion, and VM in all three human cell lines analyzed as well as in the rat glioma cells.

As far as translation into clinics is concerned, it is worth mentioning that small peptides, such as uPAcyclin, are generally non-immunogenic and cost-effective to produce, and some of them have already entered clinical trials [71]. For example, the peptide octreotide, a somatostatin agonist, has been used in combination with an mTOR inhibitor for aggressive and recurrent meningiomas where they have shown to be effective and well tolerated [72]. Nevertheless, small peptides need to reach molecular targets to accomplish their effects. Thus, they often depend on physiological transports and may be blocked by physiological barriers that prevent them from reaching therapeutic concentrations [71].

However, an alternative option to treat aggressive type of cancers is local therapy, providing a unique opportunity to deliver high doses of therapeutics to the area with the highest concentration of GBM cells, with limited systemic adverse effects. This includes treatments of the surgical cavity, novel intraoperative techniques, direct exposure of the tumor, and new means to local drug delivery under pressure and intra-arterial treatments [73].

## 5. Conclusions

This study sheds light on the effects of the uPAcyclin decapeptide, derived from a non-catalytic region of human urokinase (uPA), in one rat glioma and three human GBM cell lines. Notably, we showed that uPAcyclin inhibits migration, invasion, and formation of vascular-like structures, making it a therapeutically promising candidate for anti-GBM therapies.

## Figures and Tables

**Figure 1 cancers-15-04775-f001:**
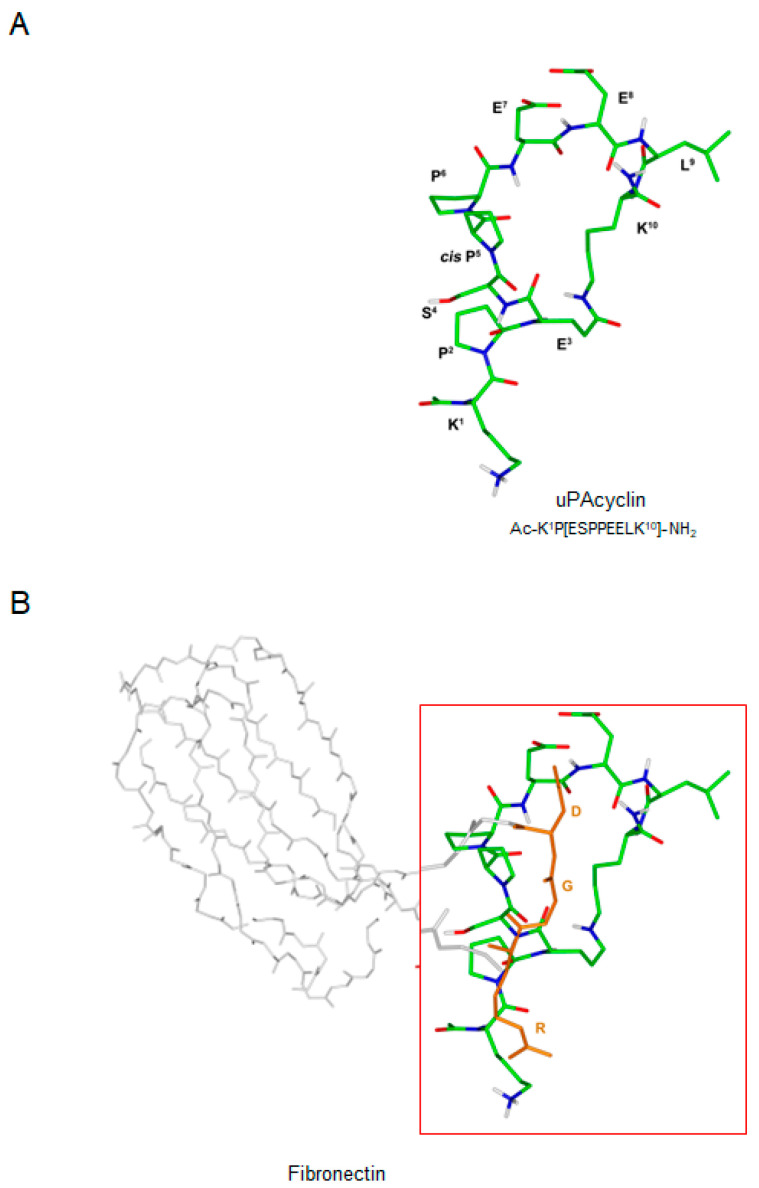
Three-dimensional structure of uPAcyclin. (**A**) Lowest energy conformer of uPAcyclin calculated using the NMR constraint reported in Appendix A through a simulated annealing procedure followed by molecular minimization. Heavy atoms are shown in different colors (carbon, grey or green; nitrogen, blue; oxygen, red; hydrogen, white). For the sake of clarity, only polar hydrogen atoms are depicted. (**B**) Superposition of the NMR structure of uPAcyclin and the crystal structure of fibronectin (pdb code 4MMX) [39]. The structures of uPAcyclin (same color codes as in (**A**)) and fibronectin (grey, only the RGD fragment is shown in orange) were superimposed using the heavy atoms in the following couples of side chains: Lys^1^/Arg and Glu^8^/Asp. For the sake of clarity, uPAcyclin and the RGD region of fibronectin (in the red box) are 10× enlarged compared to the remaining part of the protein.

**Figure 2 cancers-15-04775-f002:**
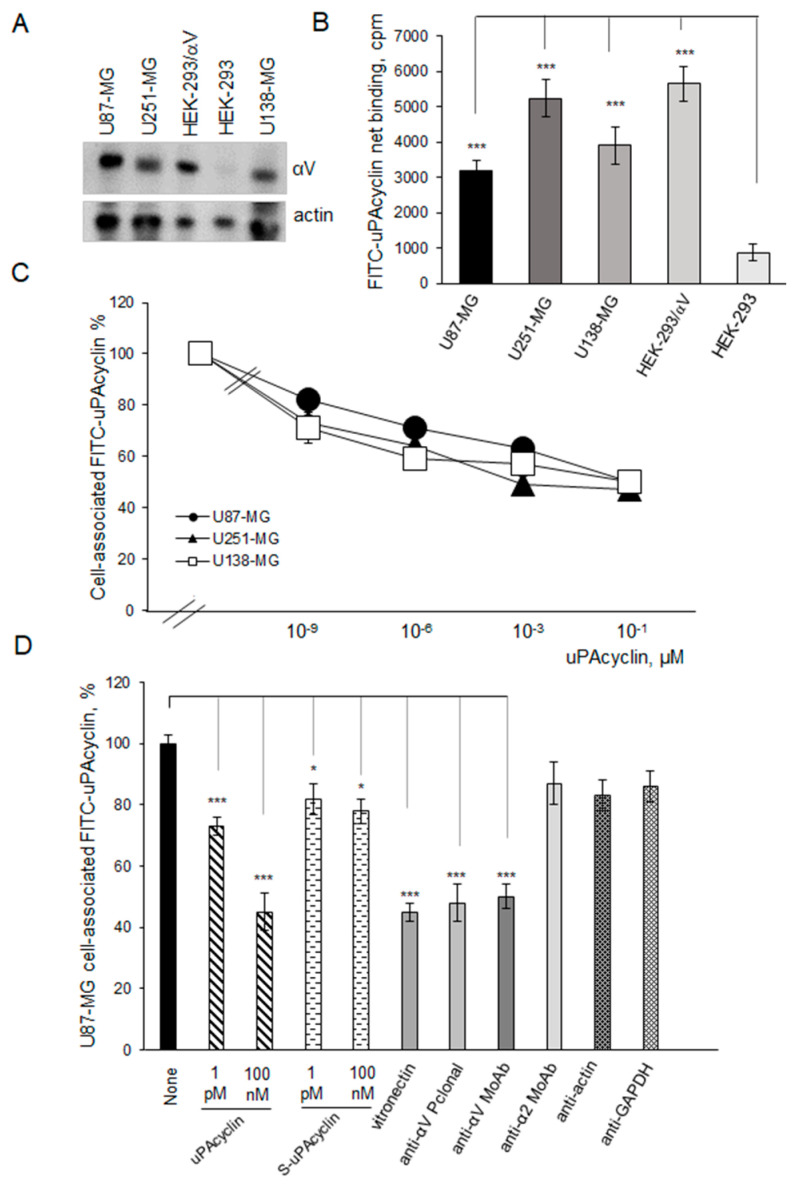
αV expression of U87-MG, U251-MG, and U138-MG GBM cells and specific binding of FITC-uPAcyclin. (**A**) U87-MG, U251-MG, U138-MG, HEK-293, and HEK-293/αV total cell lysates were subjected to Western blotting and probed using polyclonal anti-αV or polyclonal anti-actin as loading control. (**B**) A total of 2 × 10^6^ U87-MG, U251-MG, U138-MG, HEK-293/αV, or HEK-293 cells were pre-incubated for 30 min at 4 °C with 500 nM uPAcyclin and further exposed to 50 nM FITC-uPAcyclin for 2 h at 4 °C. At the end of the incubation, the unbound material was washed and cell-associated FITC-uPAcyclin was quantitated by a fluorimeter. The net value of cell-surface-associated fluorescence, corresponding to the specific binding, is shown. (**C**) A total of 2 × 10^6^ U87-MG, U251-MG, and U138-MG cells were pre-incubated for 30 min at 4 °C with increasing concentrations of unlabeled uPAcyclin and then assayed for binding of FITC-uPAcyclin, for 2 h at 4 °C. Cell-surface-associated fluorescence, as percentage of the samples without FITC-uPAcyclin, is shown. (**D**) U87-MG cells were pre-incubated with uPAcyclin (1 pM, 100 nM), scrambled uPAcyclin (S-uPAcyclin, 1 pM and 100 nM)), vitronectin (200 µM), anti-αv polyclonal (anti-αV Pclonal, 5 µg/mL) or monoclonal (anti-αV MoAb, 5 µg/mL) or anti-α2 monoclonal (anti-α2 MoAb, 1 µg/mL), polyclonal anti-actin, and monoclonal anti GAPDH. Then, cells were treated as in B (* *p*-value < 0.05; *** *p*-value < 0.001).

**Figure 3 cancers-15-04775-f003:**
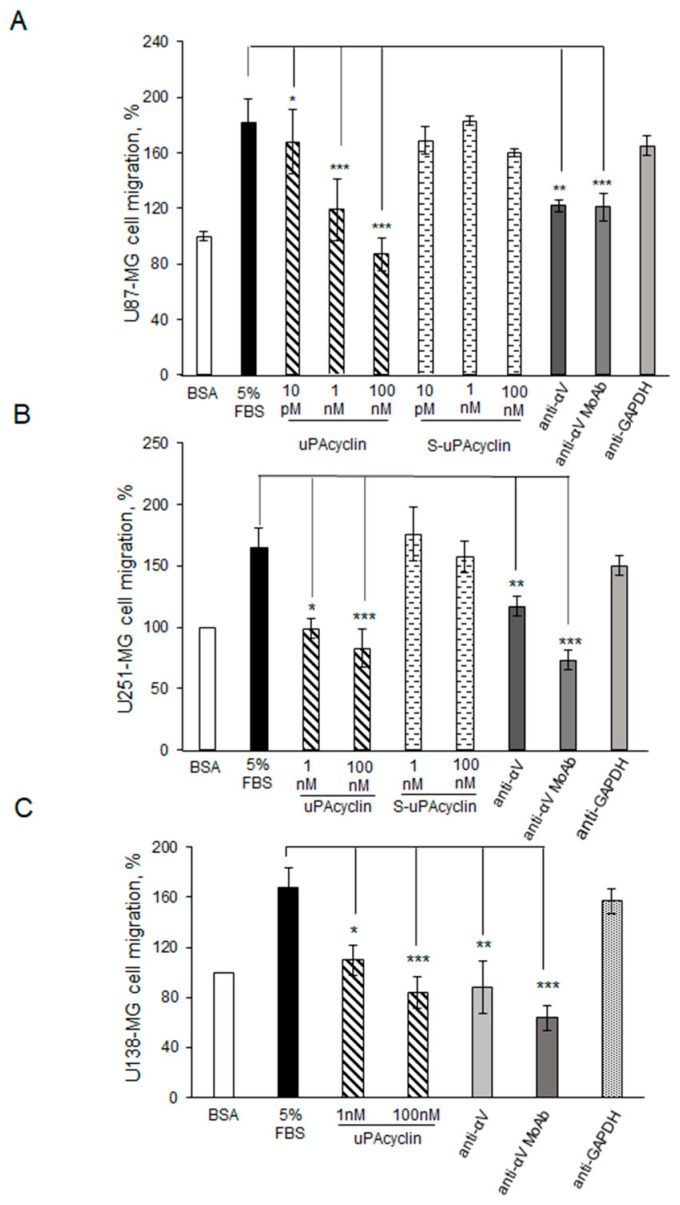
Inhibition of U87-MG, U251-MG, and U138-MG GBM cellular migration by uPAcyclin. Quantitative analysis of U87-MG (**A**), U251-MG (**B**), and U138-MG (**C**) directional migration. A total of 11 × 10^4^ U87-MG/sample or 4 × 10^4^ U251-MG or U138-MG/sample were pre-treated for 40 min at 37 °C with uPAcyclin or S-peptide at the indicated concentrations, anti-αV polyclonal (anti-αV Pclonal) or monoclonal (anti-αV MoAb) antibody, or anti-GAPDH, and anti-α2 monoclonal antibody (anti-α2 MoAb), and then assayed in Boyden chambers at 37 °C, either for basal (BSA) or directional 5% FBS-dependent migration for 3 h at 37 °C. At the end of incubation, the filters were processed as described in Methods. Basal migration was taken as 100% and all values were calculated relative to that (* *p*-value < 0.05; ** *p*-value < 0.005; *** *p*-value < 0.001).

**Figure 4 cancers-15-04775-f004:**
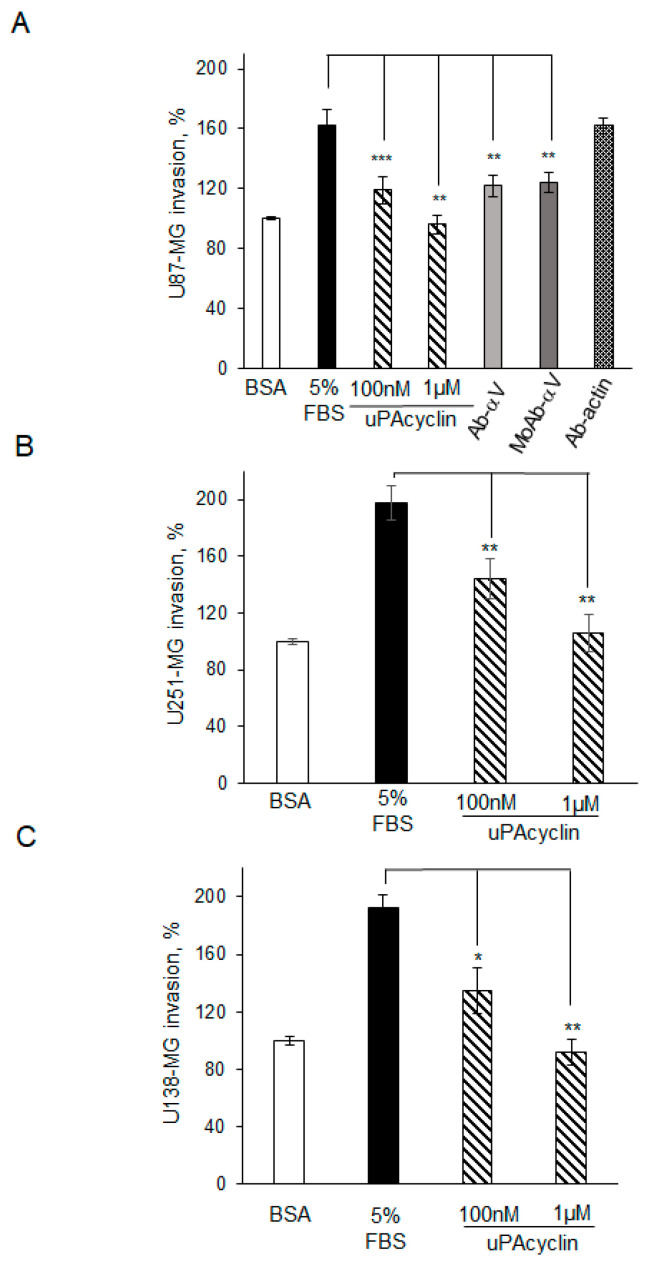
Effect of uPAcyclin on matrix invasion of U87-MG, U251-MG, and U138-MG GBM cells. Quantitative analysis of U87-MG (**A**), U251-MG (**B**), and U138-MG (**C**) invasion. A total of 11 × 10^4^ U87-MG or 4 × 104 U251-MG or U138-MG/sample were pre-treated for 40 min at 37 °C with uPAcyclin (100 nM, 1 µM), anti-αV polyclonal (anti-αV-Pclonal) or monoclonal (anti-αV MoAb) antibody, or anti-actin polyclonal antibody (anti-actin), and then assayed in Boyden chambers at 37 °C, either for basal (BSA) or FBS-dependent invasion (5% FBS) for 5 h at 37 °C. Data are expressed as in the legend to Figure 3. Results are expressed as percentages with respect to the basal invasion, considered as 100% (* *p*-value < 0.05; ** *p*-value < 0.005; *** *p*-value < 0.001).

**Figure 5 cancers-15-04775-f005:**
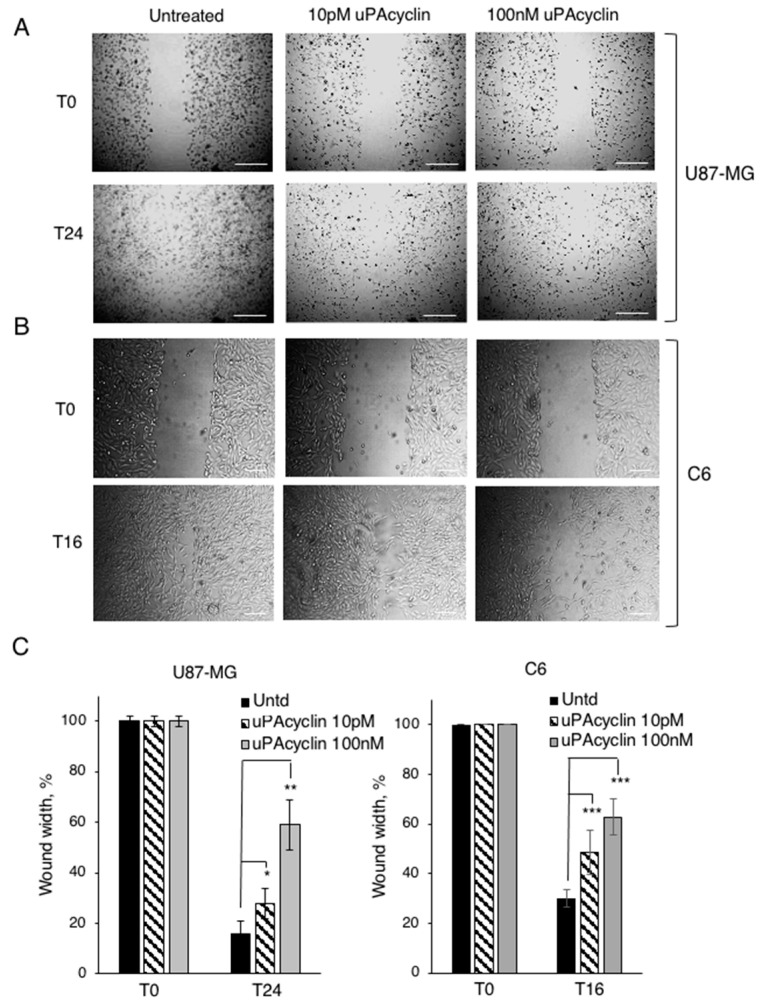
Wound healing assay of U87-MG and C6 cells, in the presence of uPAcyclin. (**A**) U87-MG cells were grown to confluence in multiwell-12 plates with inserts, pre-incubated with uPAcyclin (10 pM or 100 nM) or diluents for 40 min at 37 °C, and then exposed to 3% FBS, in the presence of uPAcyclin or diluents, as reported [25]. Images were taken at time 0 (T0) and 24 h (T24) and representative images are shown (50× total magnification, scale bar: 250 µm). (**B**) C6 cells were grown to confluence in multiwell-12 plates, wounded with a yellow tip, pre-incubated with uPAcyclin (10 pM and 100 nM) or diluents, and assayed as in panel A. Representative images of wound closure were taken at time 0 (T0) and 16 h (T16) (40× total magnification, scale bar: 100 µm). (**C**) For both U87-MG and C6, the wound width at time 0 is taken as 100% and the wound width at T24 or T16 is calculated as relative to that (* *p*-value < 0.05; ** *p*-value < 0.005; *** *p*-value < 0.001).

**Figure 6 cancers-15-04775-f006:**
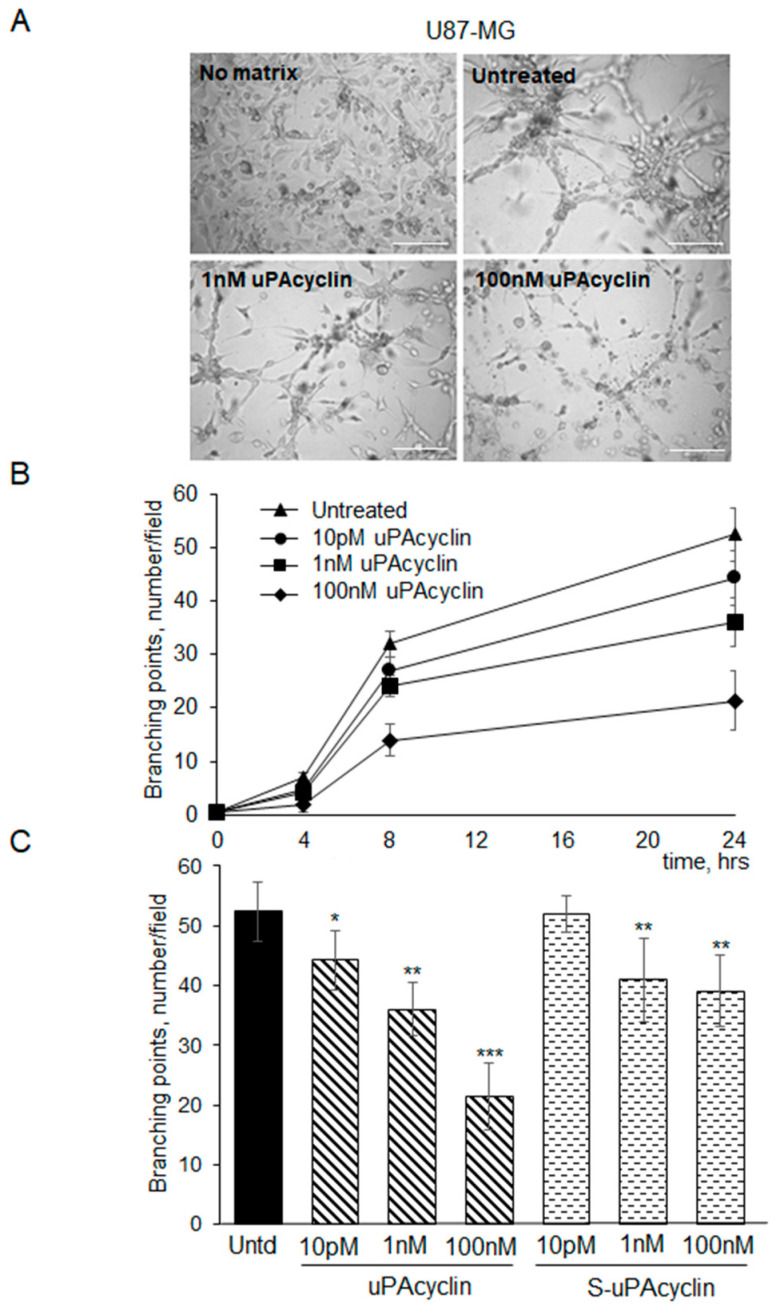
Vascular mimicry formation by U87-MG and effect of uPAcyclin. (**A**) Representative images of U87-MG cells pre-treated or not (untreated) for 40 min at 37 °C with uPAcyclin (1 nM and 100 nM) and then plated onto Geltrex^TM^ in multiwell-96 (3 × 10^4^/well). As a control, cells were grown in the presence of DMEM/10% FBS (No matrix) or with Geltrex^TM^ (Untreated, Untd). Images are taken at 24 h using the Inverted Microscope DMI Leica 6000 with 50× total magnification, scale bar: 75 µm. (**B**) The quantitation of branching points, defined as the intersection of at least three tubes [24], was performed at 4, 8, and 24 h either in control cells and cells pre-treated for 40 min at 37 °C with 10 pM, 1 nM, and 100 nM uPAcyclin before plating onto Geltrex^TM^. (**C**) U87-MG cells were pre-treated with uPAcyclin or with S-uPAcyclin, at the indicated concentrations, plated onto Geltrex^TM^ and, 24 h later, the number of branching points was evaluated and reported in the histogram (* *p*-value < 0.05; ** *p*-value < 0.005; *** *p*-value < 0.001).

**Figure 7 cancers-15-04775-f007:**
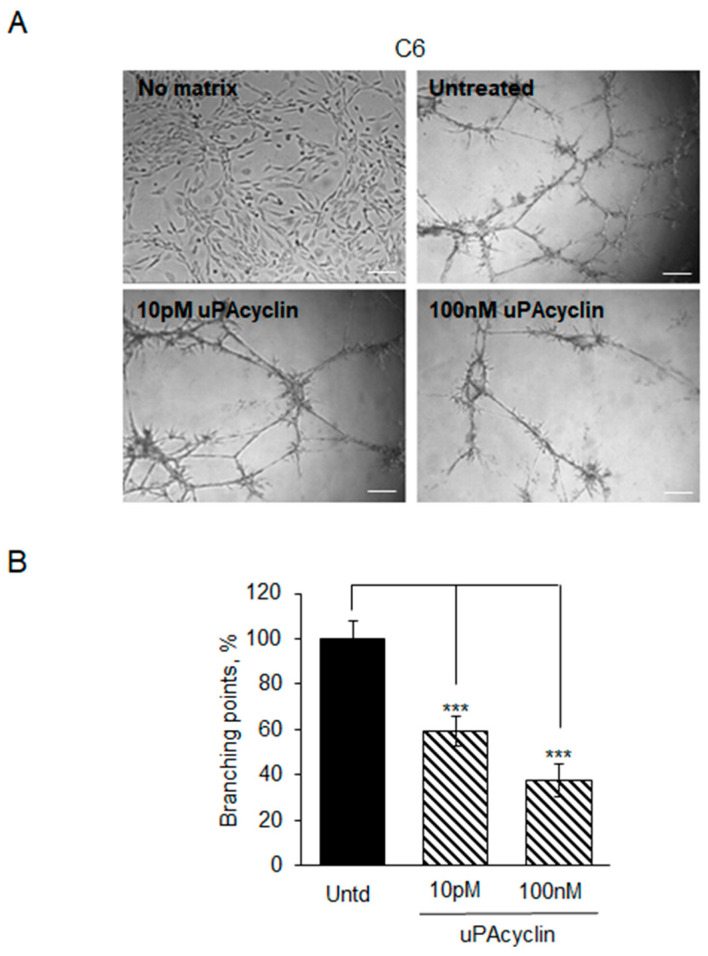
Vascular mimicry formation by C6 rat cells and inhibition by uPAcyclin. (**A**) Representative images of C6 cells, either seeded on multiwell-96 (1 × 10^4^/well), in the presence of DMEM/10% FBS (No matrix) or plated on Geltrex^TM^ (Untreated, Untd), or pre-treated with 10 pM or 100 nM uPAcyclin before seeding (40× total magnification, scale bar: 100 µm). (**B**) After 16 h, the number of branching points was evaluated as described in the legend to Figure 5 and reported in the relative histogram (*** *p*-value < 0.001).

**Figure 8 cancers-15-04775-f008:**
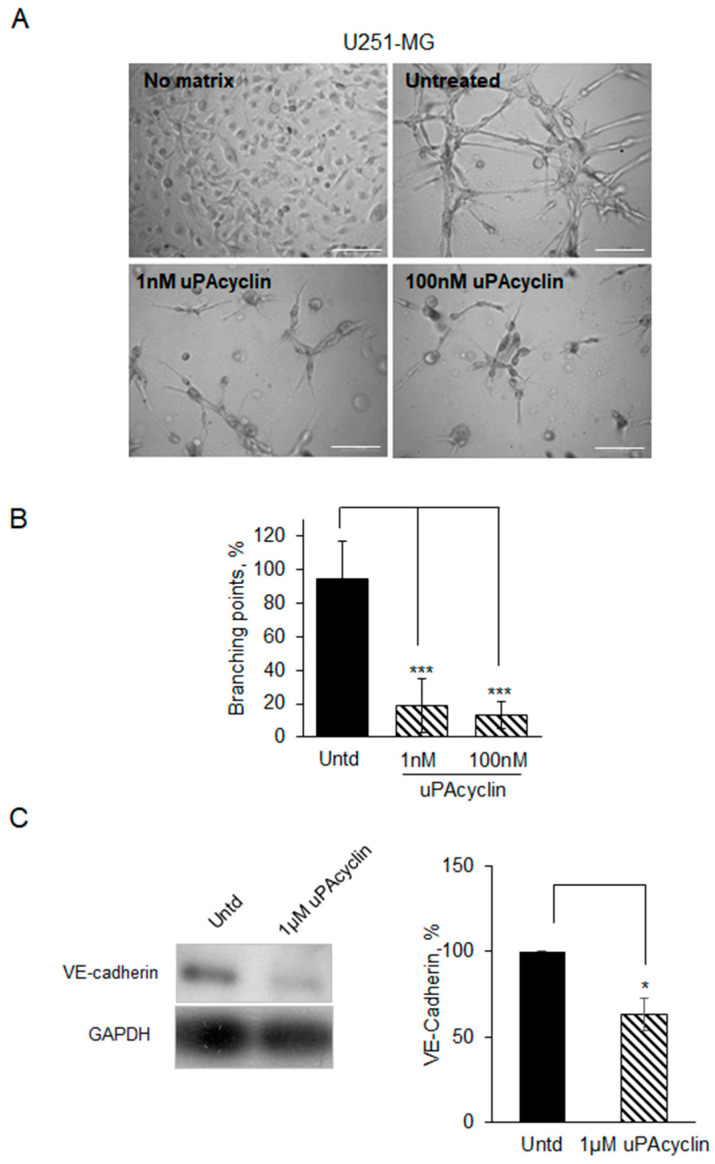
Vascular mimicry formation and VE-cadherin expression inhibited by uPAcyclin in U251-MG cells. (**A**) Representative images of U251-MG cells, either seeded in multiwell-96 (1.5 × 10^4^/well), in the presence of DMEM/10% FBS (No matrix) or plated onto Geltrex^TM^-coated, either untreated or pre-treated with 1 nM or 100 nM uPAcyclin for 24 h (50× total magnification, scale bar: 75 µm). (**B**) The quantitation of the branching point number is described in the legend to Figure 6B. (**C**) Western blotting shows the VE-cadherin content of VM-forming U251-MG cells, treated with 1 µM uPAcyclin or diluents, extracted from the matrix by enzymatic digestion and lysed as described in Materials and Methods (left panel) (Section 2). Equal amounts of cell lysates were separated on SDS-PAGE followed by Western blotting with anti-VE-cadherin or anti-GAPDH antibodies. The densitometric quantitation of VE-cadherin bands is shown (right panel) (* *p*-value < 0.05; *** *p*-value < 0.001).

## Data Availability

The data presented in this study are available in this article (and Appendix A).

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
