# Peer review of "αV-Integrin-Dependent Inhibition of Glioblastoma Cell Migration, Invasion and Vasculogenic Mimicry by the uPAcyclin Decapeptide"

_cancers, 2023, doi:10.3390/cancers15194775_

Round 1

Reviewer 1 Report

The authors present a experimental setting and tests of antiinvasiv uPAcyclin cyclic decapeptide inhibiting cell migration in human GCM cell lines. The results revealed that all CGM cells express the V-integrin subunit and bind specifically to FITC-uPAcyclin, showing a Kdapp in the low nanomolar range. Although peptide exposure neither affects viability nor cell proliferation rate, nanomolar concentrations of this uPAcyclin markedly inhibit directional migration and matrix invasion of all GBM cells. Additionally, wound healing rate closure of U87-MG and rat glioma C6 cells is reduced by 50% and the time-lapse videomicroscopy studies showed that the formation of vascular-like structures by U87-MG in three-dimensional matrix cultures is markedly inhibited by uPAcyclin. Quantitative assessments show a 50% to 80% reduction of the branching point numbers of the U87-MG, C6, and U251-MG cell lines. Lysates from matrix-recovered uPAcyclin- exposed cells exhibit a reduced expression of VE-cadherin, a prominent factor in the acquisition of vascular-like structures.

The authors concluded that their results indicate that uPAcyclin is a promising candidate to counteract the formation of new vessels in novel targeted anti-GBM therapies.

The manuscript is well written. Minor typing errors should be corrected. The methods are good and the scientific approach is well presented. The results are aequate and support the discussion and conclusion. 

I suggest to add some clinical aspects and present a link to the possible clinical setting and new therpeutical strategies in GBM patients based on these previous experimental data. 

please correct minor typing errors.

Author Response

The authors present a experimental setting and tests of antiinvasiv uPAcyclin cyclic decapeptide inhibiting cell migration in human GCM cell lines. The results revealed that all CGM cells express the V-integrin subunit and bind specifically to FITC-uPAcyclin, showing a Kdapp in the low nanomolar range. Although peptide exposure neither affects viability nor cell proliferation rate, nanomolar concentrations of this uPAcyclin markedly inhibit directional migration and matrix invasion of all GBM cells. Additionally, wound healing rate closure of U87-MG and rat glioma C6 cells is reduced by 50% and the time-lapse videomicroscopy studies showed that the formation of vascular-like structures by U87-MG in three-dimensional matrix cultures is markedly inhibited by uPAcyclin. Quantitative assessments show a 50% to 80% reduction of the branching point numbers of the U87-MG, C6, and U251-MG cell lines. Lysates from matrix-recovered uPAcyclin- exposed cells exhibit a reduced expression of VE-cadherin, a prominent factor in the acquisition of vascular-like structures.

The authors concluded that their results indicate that uPAcyclin is a promising candidate to counteract the formation of new vessels in novel targeted anti-GBM therapies. 

The manuscript is well written. Minor typing errors should be corrected. The methods are good and the scientific approach is well presented. The results are aequate and support the discussion and conclusion. 

I suggest to add some clinical aspects and present a link to the possible clinical setting and new therpeutical strategies in GBM patients based on these previous experimental data. 

Comments on the Quality of English Language

Please correct minor typing errors.

We thank this Reviewer very much for appreciating our work.

Regarding the link of uPAcyclin with clinical studies, we have now described the anti-cancer activities of A6 peptide, identified by high-throughput screening more than two decades ago. A6 partially shares the primary aminoacidic sequence with uPAcyclin and has been subjected to testing in clinical trials as anti-cancer agent. In the Discussion, lines 593-597, the following sentences with the appropriate references have been included: “The sequence of uPAcyclin shares extensive similarities with the Å6 octapeptide (Ac-K136PSSPPEE143-NH2), corresponding to uPA residues 136-143, that is endowed with remarkable anti-angiogenetic and anti-metastatic activities in mice. The Å6 was tested in gynaecologic malignancies and phase I trials indicated that it was well tolerated. Further phase II studies, indicated a delayed disease progression in treated individuals”.

The minor typing errors have been corrected.  

Reviewer 2 Report

There is a problem at line 427. Missing text?

Author Response

There is a problem at line 427. Missing text?

We thank this Reviewer for appreciating our work and for noticing some missing text. We have now included the missing four lines, as follows:

“3.5 Vascular mimicry formation by U87-MG and U251-MG glioblastoma cell lines

When U87-MG human glioblastoma cells are seeded onto culture dishes, they grow adherent to the surface without forming any particular structure (Fig 6). When cultivated under serum-free conditions in gelled extracellular matrices like GeltrexTM ,”

Reviewer 3 Report

Comments on Manuscript ID cancers-2579704:

1. Supplementary Materials are not available in the uploaded folder. Original Western blot image doesn't look convincing. Authors are recommended to send a good-quality Western blot image.

2. These findings uncover novel uPAcyclin activities and provide a strong rationale for further preclinical studies. If the findings are novel, it should not be limited to only preclinical studies. Authors need to correct the statement "......... Clinical studies."

3. Authors should add recent citations (from 2021-2023).

4. The manuscript needs grammatical corrections and better sentence structural formatting. Additionally, the language should be substantially improved before considering it for publication. Some statements appear unclear as well. Authors should seek the assistance of professional editors.

5. Discussion section, line numbers 492-493: What do you mean by "Among new therapeutic approaches for glioblastoma, peptides show the potential for more effective targeting associated with reduced side effects and drug resistance, compared to the conventional devastating anti-GBM therapies." This should be rewritten.

6. I can't see clarity in the information regarding αV-integrin and VE-cadherin. Please write this information properly. In the discussion section, on page 17, paragraph 2; the first few lines appear to reflect similar information mentioned in the 1st paragraph.

7. I sense inconsistency in the flow of the discussion.

8. Can authors justify the reason for using HEK-293/αV (HEK-293 overexpressing αV) in this study for comparison with U87-MG, U251-MG, U138-MG human glioblastoma cell lines, and C6 rat glioma? Is it the standard cell line to compare the expression of αV-integrin?

9. I can't see Western blot data of the U138-MG human glioblastoma cell lines and C6 rat glioma.

10. I can't see cell viability data, i.e., Trypan blue (Figure S1A, B, C) or MTT assays (Figure S1D, E).

11. Where did you obtain the U138-MG human glioblastoma cell lines from?

12. What were the humidity and CO2 levels during the experiments?

13. Authors should briefly describe the synthesis steps of the peptide, acetylation, and FITC-conjugation process in the supplementary file.

14. What is the significance of the wound healing assay in this study?

15. Fig 6A: Images are taken at different magnifications. Please provide images at the same magnification and make the scale visible in this image.

16. Why are U87-MG, U251, and C6 cells not at the same confluency for the vasculogenic mimicry assay? Why are their evaluation times different? Please provide the doubling time and growth rate of these cell lines. Authors have mentioned U138-MG human glioblastoma cell lines in their manuscript; where did they use this cell line?

17. What was the total magnification at which the cells were evaluated? What were the magnifications of the objective lens and ocular lens?

18. Regarding cell recovery from the 3D matrix: Lysates were centrifuged at 120,000 rpm. The speed of 120,000 rpm is extremely high and may not be practical in most laboratory settings. Is this the correct speed, and which centrifuge did you use?

19. Please provide the raw data of your replication (n=3) experiments.

20. What is the RGD fragment? Define it in the text.

21. Section 3.2: The first 3 lines look unnecessary. Please remove them. Why was this section given such a title, "Specific binding of FITC-uPAcyclin to U87-MG glioblastoma cells," when FITC-uPAcyclin has affinity for both U87-MG and U251-MG cells? What about U138-MG human glioblastoma and C6 rat glioma cell lines? Why did the authors not test binding affinity on these cell lines?

22. 3.3. Inhibition of U87-MG, U251-MG, and U138 GBM cell migration and invasion by uPAcyclin: Why did authors not test S-uPAcyclin at 100 nM concentration for cell migration assay for U87-MG cell line and not at all test for U251-MG and U138-MG cell lines for this test and invasion test? What is the logic of using anti-alpha2 MoAb?

23. What is Kdₐpp?

24. Fig 5B: Stretching is not uniform at T0. At T16, I can't see much difference between untreated and 100 µM uPAcyclin. Again, at T24, there is no significant change between 10 µM and 100 µM uPAcyclin.

25. Fig 6A and B, 7A and B, Fig 8: Where is the data for S-uPAcyclin?

26. What does Fig 2D indicate? Please explain it properly.

27. Please write the discussion section in line with your findings.

Improve language to increase the readability. 

Author Response

1.Supplementary Materials are not available in the uploaded folder. Original Western blot image doesn't look convincing. Authors are recommended to send a good-quality Western blot image.

Now, Figure 2, panel A has been substituted to include the alphaV detection in the U138-MG lysates and the new original blot is uploaded in the Supplementary Materials. The original blot shown in Figure 8 is also included (Figure 2-8: original blots).

  1. These findings uncover novel uPAcyclin activities and provide a strong rationale for further preclinical studies. If the findings are novel, it should not be limited to only preclinical studies. Authors need to correct the statement "......... Clinical studies."

The sentence has been corrected as requested: “These findings uncover novel uPAcyclin activities and provide a strong rationale for further clinical studies.”

  1. Authors should add recent citations (from 2021-2023).

We have added several new references included in the suggested timeframe. The oldest refer to methodologies or to seminal work on the subject. Now, about 30% of the total citations are relative to the last five years.

  1. The manuscript needs grammatical corrections and better sentence structural formatting. Additionally, the language should be substantially improved before considering it for publication. Some statements appear unclear as well. Authors should seek the assistance of professional editors.

The manuscript has now been reviewed thoroughly and language has been improved by the aid of an English-speaking colleague.

  1. Discussion section, line numbers 492-493: What do you mean by "Among new therapeutic approaches for glioblastoma, peptides show the potential for more effective targeting associated with reduced side effects and drug resistance, compared to the conventional devastating anti-GBM therapies." This should be rewritten.

The sentence has been re-written: “Among new anti-cancer therapeutics, peptides show the potential for an effective targeting associated to reduced side effects and drug resistance”. To substantiate this statement, a recent review regarding “Peptides as multifunctional players in cancer therapy” by Vadevoo et al., has been included”(2023).

  1. I can't see clarity in the information regarding αV-integrin and VE-cadherin. Please write this information properly. In the discussion section, on page 17, paragraph 2; the first few lines appear to reflect similar information mentioned in the 1st paragraph.

The paragraph describing (lines 483-495) the evaluation of VE-cadherin in U251 cells undergoing VM (shown in Figure 8) was re-written to clarify all information about the rationale, procedure and results. In the Discussion the first few lines of paragraph 2 have been rephrased, eliminating redundant information (lines 578-580).

  1. I sense inconsistency in the flow of the discussion.

The Discussion has been thoroughly revised.

  1. Can authors justify the reason for using HEK-293/αV (HEK-293 overexpressing αV) in this study for comparison with U87-MG, U251-MG, U138-MG human glioblastoma cell lines, and C6 rat glioma? Is it the standard cell line to compare the expression of αV-integrin?

HEK-293/αV (HEK-293 overexpressing αV) is a in-house stable clone producing 3-4 fold more αV than the parental cells (published in Belli et al, 2020). We employed it as a control for alphav detection in WB and receptor binding assays. Indeed, the aim of this experiment is to assess the occurrence of alphav (peptide target) in the U87-MG, U251-MG and U138-MG cell lysates, before proceeding with further studies..

  1. I can't see Western blot data of the U138-MG human glioblastoma cell lines and C6 rat glioma.

In this study, we mainly focus on the human GBM cell lines. The C6 rat cell line is employed only in the wound healing and VM tests and we have not searched for aV in C6 lysates. Given the short time allowed for revisions, we could not perform this experiment. However, we have now included the probing of U138-MG lysates with anti-aV antibodies, as requested. As specified in the response to criticism n.1, Figure 2, panel A shows the aV expression of all three human GBM cell lines employed in this study.

  1. I can't see cell viability data, i.e., Trypan blue (Figure S1A, B, C) or MTT assays (Figure S1D, E).

In the Supplementary Figure 1, uploaded as Supplementary material, the number of viable cells in the presence or absence of 1nM or 100nM uPAcyclin was evaluated as a function of time for 72-96 hrs. U87-MG (A), U251-MG (B) and U138-MG (C) cells did not show any significant peptide-dependent changes in the resulting curve trend.

Cell viability of U87-MG and C6, tested by MTT assay (D, E) in the presence or in the absence of 1nM or 100nM uPAcyclin for 24 hrs, did not show any difference.

  1. Where did you obtain the U138-MG human glioblastoma cell lines from?

As now stated in the 2.1 Cell lines and culture conditions section: “…U138-MG cells were obtained from the ICLC, Interlab Cell Line Collection (National Institute for Cancer Research Genoa, Italy).”

  1. What were the humidity and CO2 levels during the experiments?

During the biological assays (migration, invasion, wound healing and vasculogenic mimicry) samples were kept in the incubators for cell cultures, ensuring an approx 95% humidity and 5% CO2.

  1. Authors should briefly describe the synthesis steps of the peptide, acetylation, and FITC-conjugation process in the supplementary file.

These procedures were accurately described and included in the Supplementary information.

  1. What is the significance of the wound healing assay in this study?

The wound healing assay measures cell random migration, in the absence of motogen factors. In contrast,  the Boyden chamber assay, evaluates directional migration (toward serum). In this respect, the two assays provide different, but complementary information.

  1. Fig 6A: Images are taken at different magnifications. Please provide images at the same magnification and make the scale visible in this image.

As specified in the Figure legend, scale bar is 75µm in all images and the microscopy observations were conducted in the same way for all of them. However, cells may appear different in size, as control cells, in the absence of Geltrex matrix, are morphologically different from the elongated, well connected, vascular-like shaped U87-MG cells.

  1. Why are U87-MG, U251, and C6 cells not at the same confluency for the vasculogenic mimicry assay? Why are their evaluation times different? Please provide the doubling time and growth rate of these cell lines. Authors have mentioned U138-MG human glioblastoma cell lines in their manuscript; where did they use this cell line?

-This is correct. As specified in Methods, 3x104 U87-MG, 1.5x104U251 and 1x104 C6 cells /100µl/well are plated onto GeltrexTM for the VM assay. Also, U87 and U251 were monitored for 24 hrs, whereas C6 for 16 hrs. These conditions, taking into account the different growth rates and VM formation, resulted from our pilot studies to find the best conditions for monitoring and quantitating the vascular-like structure formation by all cells.

-The doubling time and growth rate of these cell lines are included in the Figure S1, in the supplementary files uploaded with the manuscript. Please, cfr also response n.10.

-The U138-MG cells were employed as a third GBM cell line not undergoing VM, but still sensitive to the anti-migratory and and-invasive properties of uPAcyclin. A new sentence has been included in the Discussion, lines 536-538: ”It is noteworthy that the inhibitory effect of uPAcyclin is exerted also on the aV-expressing U138-MG cell line, not undergoing VM, but sensitive to the anti-migratory and anti-invasive effect of the decapeptide.”

  1. What was the total magnification at which the cells were evaluated? What were the magnifications of the objective lens and ocular lens?

For wound healing and VM assays, we employed the Objective 5x N plan 5x/0.12 and the Eyepiece Leica HC plan 10x/25, so that the total magnification of the objective lens combined with the eyepiece is equal to 50x magnification.

  1. Regarding cell recovery from the 3D matrix: Lysates were centrifuged at 120,000 rpm. The speed of 120,000 rpm is extremely high and may not be practical in most laboratory settings. Is this the correct speed, and which centrifuge did you use?

We are sorry, there was a typo in the text, the lysates were centrifuged in a microfuge, at 12,000 rpm for 15 minutes (line 270).

  1. Please provide the raw data of your replication (n=3) experiments.

We uploaded the excel data of migration, invasion and binding assays experiments for all cell lines. Specific sentences about the number of replicates have been introduced in Methods, paragraph “2.5 Binding assay” and “2.7 Migration and invasion assays”.

  1. What is the RGD fragment? Define it in the text.

The RGD is now described at first mention in the text (Line 307-308).

  1. Section 3.2: The first 3 lines look unnecessary. Please remove them. Why was this section given such a title, "Specific binding of FITC-uPAcyclin to U87-MG glioblastoma cells," when FITC-uPAcyclin has affinity for both U87-MG and U251-MG cells? What about U138-MG human glioblastoma and C6 rat glioma cell lines? Why did the authors not test binding affinity on these cell lines?

We agree with these observations:

-The first unnecessary three lines were removed.

-Concerning C6 cells, please refer to the response n.9.

-In the revised version, we included the FITC-uPAcyclin binding assay to U138-MG cells (new Figure 2C).

-Therefore, the title of this paragraph is as follows: “3.2. Specific binding of FITC-uPAcyclin to U87-MG, U251-MG and U138-MG GBM cells”.

  1. 3.3. Inhibition of U87-MG, U251-MG, and U138 GBM cell migration and invasion by uPAcyclin: Why did authors not test S-uPAcyclin at 100 nM concentration for cell migration assay for U87-MG cell line and not at all test for U251-MG and U138-MG cell lines for this test and invasion test? What is the logic of using anti-alpha2 MoAb?

-The uPAcyclin decapeptide, its binding site, specificity and biological activity in breast cancer were previously published by this group (Belli et al., 2020). Here, in the revised version, we show one cyclic control peptide specifically synthesized for negative control of uPAcyclin: the scrambled uPAcyclin, carrying the same aminoacidic residues in a scrambled order (S-uPAcyclin).

This control is now shown in:

-the new Figure 2D, in the FITC-uPAcyclin binding assay to the U87-MG cells: S-uPAcyclin, 1 pM and 100 nM.

-the new Figure 3, in migration tests:

Panel A: S-uPAcyclin, 1 pM, 1 nM and 100 nM to U87-MG cells

Panel B: S-uPAcyclin, 1 nM and 100 nM to U251 cells.

The following new statements are included in Results:

-Concerning the binding assay, lines 328-330: “.. S-uPAcyclin retains very little binding ability to U87-MG cells (Figure 2D). This may be due to the persisting interactions of critical aminoacids shared between the sequences of uPAcyclin and S-uPAcyclin.”

-Concerning the migration assay, lines 370-371: “In this assay, the S-uPAcyclin peptide is ineffective up to 100nM (Figure 3A, B).

-Anti a2 MoAb were used as a negative control, because we know from previous experiments (Belli et al, 2020) that uPAcyclin, even at high concentration, does not bind to a2 integrin.

  1. What is Kdₐpp?

In the binding assay shown in Figure 2C, the Kdapp is the concentration of the unlabeled ligand at which half of the ligand binding sites (aV) are occupied by the FITC-uPAcyclin. The smaller the Kdapp value, the greater the binding affinity of the ligand for its target.

  1. Fig 5B: Stretching is not uniform at T0. At T16, I can't see much difference between untreated and 100 µM uPAcyclin. Again, at T24, there is no significant change between 10 µM and 100 µM uPAcyclin.

This assay is employed for assessing relative random migration and quantitated by analysing the wound images at the indicated times. We agree that it is hard to judge simply looking at the images. For this reason,  we developed a simple methodology, that is now described in the 2.8 Wound Healing Assay paragraph, as follows: “To quantitate cell migration, a rectangle covering the wound edges is drawn on the T0 image and is further applied to the single photograms of each sample. The average margin distance is then measured in the same three points and the resulting data expressed as percentage wound width of the T0 distance, taken as 100%.

  1. Fig 6A and B, 7A and B, Fig 8: Where is the data for S-uPAcyclin?

As described in response n. 22, the uPAcyclin decapeptide, its binding site, specificity and biological activity in breast cancer were previously published by this group (Belli et al., 2020). In the revised version, the scrambled uPAcyclin (S-uPAcyclin) was tested in binding assays with FITC-uPAcyclin and in the migration/invasion assays. Although S-uPAcyclin retains very little binding ability (possibly due to the interactions of single aminoacidic residues with the target), does not exert inhibitory effects at the concentrations tested, up to 100 nM.

  1. What does Fig 2D indicate? Please explain it properly.

Because Figure 2 has been substituted with a new Figure 2, the description of the results in the “3.2. Specific binding of FITC-uPAcyclin to U87-MG, U251-MG and U138-MG GBM cells” paragraph and Figure legend were changed accordingly.

  1. Please write the discussion section in line with your findings.

The Discussion paragraph had been extensively revised.

Comments on the Quality of English Language

Improve language to increase the readability. 

The manuscript has now been reviewed thoroughly and language has been improved by the aid of an English-speaking colleague.

Reviewer 4 Report

The authors of this work evaluated the uPAcyclin decapeptide in cell migration, invasion and vascular mimicry formation on human glioblastoma cells. The work shows many experiments evaluating cellular effects on migration, invasion and vascular mimicry. There are few investigations related to the mechanisms which could have increased the scientific significance of the work.

Main comments:

#1. I suggest using less technical terms in the abstract (i.e. FITC-uPAcyclin, Kdapp ) and keeping the language focused on the rationale, aims and general results of the work.

#2. I suggest deleting the term "multiforme", since it is obsolete today. It is no longer indicated in the most recent WHO classification.  However, GBM abbreviation can be used for GlioBlastoMa as authors reported in the main text correctly. Please, check the reference below:

Louis DN, et al. The 2021 WHO Classification of Tumors of the Central Nervous System: a summary. Neuro Oncol. 2021 Aug 2;23(8):1231-1251. doi: 10.1093/neuonc/noab106. PMID: 34185076; PMCID: PMC8328013.

#3. In the abstract the authors initially mention three cell lines ( U87-MG, U251-MG and U138-MG)  but then the rat glioma C6 cells line also appears. This part needs to be corrected.

#4. The authors should extensively introduce the process of vascular mimicry, both in general and in the context of GBM.

#5. The same goes for integrins. It is necessary to introduce their mechanism of action in the processes of invasion.

#6. Lines 125-126: “Using several human and rat cell lines (U87-MG, 125 U251-MG, U138-MG GBM and C6 rat glioma cells)”. Only one rat glioma cell line was used, not several.

#7. The description on uPAcyclin decapeptide and urokinase is convoluted and complicated for the reader to follow. I suggest to the authors to insert an introductory figure to understand well the structure and function of both, possibly related to processes regulated by integrins and associated to VM. Alternatively, it is necessary to explain this part better and in detail.

#8. After reading the simple summary, abstract and introduction, a general confusion about the objectives of the work emerges in the reader. The introduction is long and full of concepts not well linked. The reader loses sight of the goal of the work.

#9. It is not clear why the authors decide to evaluate VE-cadherin expression only in U251. This raises questions about the evaluation of the mechanisms that would explain the effects observed by the experiments.

Minor:

- I suggest removing abbreviations from the abstract and simple summary.

- Image captions are mixed in with the main text.

The grammatical structure of some sentences could be improved.

Author Response

The authors of this work evaluated the uPAcyclin decapeptide in cell migration, invasion and vascular mimicry formation on human glioblastoma cells. The work shows many experiments evaluating cellular effects on migration, invasion and vascular mimicry. There are few investigations related to the mechanisms which could have increased the scientific significance of the work.

Main comments:

#1. I suggest using less technical terms in the abstract (i.e. FITC-uPAcyclin, Kdapp ) and keeping the language focused on the rationale, aims and general results of the work.

In the abstract and in the simple summary, we avoided abbreviations, with the exception of GBM.

In the abstract, the following sentence:”…and bind specifically to FITC-uPAcyclin, showing a Kdapp in the low nanomolar range.” was substituted with “…and bind specifically to nanomolar concentrations of uPAcyclin”.

#2. I suggest deleting the term "multiforme", since it is obsolete today. It is no longer indicated in the most recent WHO classification.  However, GBM abbreviation can be used for GlioBlastoMa as authors reported in the main text correctly. Please, check the reference below:

Louis DN, et al. The 2021 WHO Classification of Tumors of the Central Nervous System: a summary. Neuro Oncol. 2021 Aug 2;23(8):1231-1251. doi: 10.1093/neuonc/noab106. PMID: 34185076; PMCID: PMC8328013.

The term “multiforme” has now been removed from the text. The suggested reference (Luis DN et al., 2021) is now cited in the manuscript.

#3. In the abstract the authors initially mention three cell lines (U87-MG, U251-MG and U138-MG)  but then the rat glioma C6 cells line also appears. This part needs to be corrected.

The C6 cell line is now cited at the beginning of the abstract, together with the three human GBM cell lines.

At the end of Introduction, the following sentence has been included:” Using four cell lines (U87-MG, U251-MG, U138-MG human GBM and C6 rat glioma cells)...”

#4. The authors should extensively introduce the process of vascular mimicry, both in general and in the context of GBM.

The lines 55-75 of the Introduction are now devoted to the description of VM in general and in the context of GBM. The following reference has been included: Maddison, K., Bowden, N.A., Graves, M.C. et al. Characteristics of vasculogenic mimicry and tumour to endothelial transdifferentiation in human glioblastoma: a systematic review. BMC Cancer 23, 185 (2023).

#5. The same goes for integrins. It is necessary to introduce their mechanism of action in the processes of invasion.

The lines 82-99 of the Introduction now describe the integrins and their role in GBM migration and invasion. The following reference is now included: “Che P, Yu L, Friedman GK, Wang M, Ke X, Wang H, Zhang W, Nabors B, Ding Q, Han X. Integrin αvβ3 Engagement Regulates Glucose Metabolism and Migration through Focal Adhesion Kinase (FAK) and Protein Arginine Methyltransferase 5 (PRMT5) in Glioblastoma Cells. Cancers (Basel). 2021 Mar 5;13(5):1111. doi: 10.3390/cancers13051111. PMID: 33807786; PMCID: PMC7961489.

#6. Lines 125-126: “Using several human and rat cell lines (U87-MG, 125 U251-MG, U138-MG GBM and C6 rat glioma cells)”. Only one rat glioma cell line was used, not several. 

This mistake is now corrected: “Moreover, using four cell lines (U87-MG, U251-MG, U138-MG human GBM and C6 rat glioma cells), we have evaluated the ability of this cyclic decapeptide to inhibit migration and invasion, as well as the formation of vascular-like structures.” (lines 128-129)

#7. The description on uPAcyclin decapeptide and urokinase is convoluted and complicated for the reader to follow. I suggest to the authors to insert an introductory figure to understand well the structure and function of both, possibly related to processes regulated by integrins and associated to VM. Alternatively, it is necessary to explain this part better and in detail.

Lines 110-117. The text has been shortened and simplified to meet the reviewer’s criticism.

#8. After reading the simple summary, abstract and introduction, a general confusion about the objectives of the work emerges in the reader. The introduction is long and full of concepts not well linked. The reader loses sight of the goal of the work.

The Introduction has now been re-written, simplifying the concepts and exposing them in a logical order.

#9. It is not clear why the authors decide to evaluate VE-cadherin expression only in U251. This raises questions about the evaluation of the mechanisms that would explain the effects observed by the experiments.

To evaluate the level of VE-cadherin in vascular-like structures, we could not rely on a published procedure. Rather, we had to set up a new procedure for gentle extraction of cells from the Geltrex matrix and subsequent lysates preparation and western blotting analysis. To this end, different mixtures of enzymes were tried to find the right combination working on U251 cells. Repeating the full experiment and the extraction procedure on U87-MG cells would take more time than allowed for revisions. However, because the timing and type of structures formed by the two cell lines are similar, we are confident that the results obtained for U251 cells can be extended to U87-MG cells.

Minor:

- I suggest removing abbreviations from the abstract and simple summary.

All abbreviations have been removed from the abstract and simple summary except glioblastoma (GBM).

- Image captions are mixed in with the main text.

Image captions have been correctly spaced.

 The grammatical structure of some sentences could be improved.

The paper has been further reviewed.

Round 2

Reviewer 3 Report

Dear Authors,

I can't see supplementary folder having good quality western blot image as replied by you. You have simply uploaded previous western blot image labeled as fig 8 in pdf format. Further I cant see doubling time and growth rate data which is mentioned by you in the reply "In the Supplementary Figure 1, uploaded as Supplementary material, the number of viable cells in the presence or absence of 1nM or 100nM uPAcyclin was evaluated as a function of time for 72-96 hrs." 

Fig 5 A and B, 6A, 7A scale bar is not visible perfectly. For Fig 5A and B and 6A as you replied total magnification was 50x why its not corrected in text?

No scale bar information and magnification information is mentioned in fig 8A.  

Please resolve these issues.

Thanks 

Author Response

I can't see supplementary folder having good quality western blot image as replied by you. You have simply uploaded previous western blot image labeled as fig 8 in pdf format.

There must have been a mistake or a misunderstanding. We have uploaded a .pdf file with the original western blotting shown in Figure 2 and in Figure 8. The images are now pasted below.

Further I cant see doubling time and growth rate data which is mentioned by you in the reply "In the Supplementary Figure 1, uploaded as Supplementary material, the number of viable cells in the presence or absence of 1nM or 100nM uPAcyclin was evaluated as a function of time for 72-96 hrs." 

Doubling times and growth rates were not calculated, as growth curves of GBM cells, in the presence and in the absence of uPAcyclin, were almost superimposable. This well agrees with previous data published using breast cancer cells (Belli et al., 2020). However, the following sentence is now included: “The growth rates of all three cell lines were similar, regardless of the exposure to 1nM or 100nM uPAcyclin (lines 403-405)”.

Fig 5 A and B, 6A, 7A scale bar is not visible perfectly. For Fig 5A and B and 6A as you replied total magnification was 50x why its not corrected in text?

We agree with this Reviewer, the scale bar for Fig 5 A and B, 6A, 7A is barely visible, therefore it has now been corrected graphically. Total magnification is now included in the relative Figure legends.

No scale bar information and magnification information is mentioned in fig 8A.  

In Fig 8A, the scale bar was barely visible and now has been graphically corrected. Scale bar information and magnification information is now included in Figure 8 legend.

Please resolve these issues.

We thank this Reviewer for the thorough, accurate and constructive revision of our manuscript.
